# Ultra-Processed Food Intake as an Effect Modifier in the Association Between Depression and Diabetes in Brazil: A Cross-Sectional Study

**DOI:** 10.3390/nu17152454

**Published:** 2025-07-28

**Authors:** Yunxiang Sun, Poliana E. Correia, Paula P. Teixeira, Bernardo F. Spiazzi, Elisa Brietzke, Mariana P. Socal, Fernando Gerchman

**Affiliations:** 1Department of Health Policy and Management, Johns Hopkins Bloomberg School of Public Health, Baltimore, MD 21202, USA; msocal1@jhu.edu; 2Faculdade de Medicina, Universidade Federal do Rio Grande do Sul (UFRGS), Porto Alegre 90035-003, RS, Brazil; pcorreia@hcpa.edu.br (P.E.C.); papteixeira@hcpa.edu.br (P.P.T.); bspiazzi@hcpa.edu.br (B.F.S.); 3Department of Psychiatry, Queens University School of Medicine, Kingston, ON K7L 2V7, Canada; elisa.brietzke@queensu.ca; 4Division of Endocrinology and Metabolism, Hospital de Clínicas de Porto Alegre, Porto Alegre 90035-903, RS, Brazil; 5Graduate Program in Medical Sciences, Endocrinology, Department of Internal Medicine, Faculdade de Medicina, Universidade Federal do Rio Grande do Sul (UFRGS), Porto Alegre 90035-003, RS, Brazil

**Keywords:** brazilian population, depression, food choice, nutritional epidemiology, psychiatric comorbidity

## Abstract

**Background/Objectives:** Recent studies linked a diet rich in ultra-processed foods (UPFs) with depression and diabetes. Although common risk factors, such as aging, are defined for both diseases, how UPFs are associated with the bidirectional relationship between them is not known. This study aimed to investigate whether UPF intake modifies the association between depression and diabetes within the Brazilian adult population. **Methods:** This cross-sectional analysis utilized data from the 2019 Brazilian National Health Survey, involving over 87,000 adults (aged 18–92 years). Participants provided self-reported data on diabetes and depression diagnoses, dietary habits (assessed by qualitative FFQ), as well as demographic, and socioeconomic variables. Multivariate logistic regression models were used to evaluate the associations, employing two classification methods—UPF1 and UPF2—based on different thresholds of weekly consumption, for high/low UPF intake. Analyses were stratified by age groups to identify variations in associations. **Results:** There was a significant association between depression and diabetes, especially among participants with high UPF consumption. Models adjusted by demographic characteristics, as well as meat and vegetable consumptions, demonstrated elevated odds ratios (ORs) for diabetes among individuals with depression consuming high levels of UPF, compared to those with a low UPF intake (OR: 1.258; 95% CI: 1.064–1.489 for UPF1 and OR: 1.251; 95% CI: 1.059–1.478 for UPF2). Stratified analysis by age further amplified these findings, with younger individuals showing notably stronger associations (non-old adult group OR: 1.596; 95% CI: 1.127–2.260 for UPF1, and OR: 6.726; 95% CI: 2.625–17.233 for UPF2). **Conclusions:** These findings suggest that high UPF intake may influence the relationship between depression and diabetes, especially in younger adults. Future longitudinal studies are warranted to establish causality, investigate underlying biological mechanisms, and examine whether improving overall nutrient intake through dietary interventions can reduce the co-occurrence of depression and diabetes.

## 1. Introduction

Major depressive disorder is a mental health condition characterized by persistent low mood and loss of interest in previously enjoyable activities [1]. In the aftermath of the COVID-19 pandemic, depression has gained increasing attention, with an estimated 53 million cases of major depression attributable to COVID-19, and the worldwide prevalence of depression in 2021 surged to 246 million [2]. In Brazil, the lifetime prevalence of major depressive disorder was found to be 10%, which is in the top three among developing countries. It is currently the second leading cause of disability in the country [3,4]. Depression is also projected to become the primary contributor to the global disease burden by 2030 [5]. Middle-aged and older adults bear a more significant burden due to depression, as it negatively affects their psychosocial well-being, impairs daily functioning, and reduces their overall quality of life [6].

Additionally, depression is particularly associated with a significant decline in overall health when combined with diabetes [7]. Individuals with both conditions have a significantly higher likelihood of functional disability—2.95 times greater than those with diabetes alone and 2.38 times greater than those with depression alone [8]. Recent data has found no shared genetic factors explaining the association between depression and diabetes, while multiple environmental risk factors—particularly related to socioeconomic status, education, and lifestyle—are common to both conditions [9]. In fact, in a recent systematic review and meta-analysis, we have shown that lifestyle interventions were effective at improving the severity of depressive symptoms in individuals with diabetes [10]. However, currently, studies on the relationship between diabetes and depression mainly focus on the common etiology and the same health outcomes, with limited exploration of how dietary patterns might modify the relationship between these two conditions [8].

Ultra-processed foods (UPFs), as defined by the NOVA classification, are industrial formulations typically containing additives, colorants, and emulsifiers not commonly found in home cooking and are widely recognized as unhealthy due to their association with poor dietary quality and adverse health outcomes [11,12,13]. Research consistently shows that UPF consumption is associated with metabolic conditions such as obesity, diabetes, and cardiovascular diseases. UPF consumption is also associated with emotional eating patterns linked to depression, which can lead to increased UPF intake [14]. Large-scale studies from the United States, Italy, and South Korea have demonstrated consistent associations between high UPF intake and both mental and metabolic disorders [15,16,17]. However, research in developing countries remains scarce, where the obesity and diabetes epidemics are rising [18]. The high intake of calorie-dense, nutrient-poor UPFs, which are rich in refined carbohydrates and saturated fats, has been well-documented as a risk factor for diabetes and obesity [19].

Age may play an important role in moderating the effects of UPF consumption on health outcomes. Younger individuals, particularly adolescents and young adults, tend to consume more UPFs compared to older adults due to factors such as food marketing exposure, convenience, lower cost, and less health-conscious dietary habits [20,21]. Studies have shown that dietary behaviors established early in life often persist into adulthood, potentially compounding long-term health risks [22]. Moreover, the physiological and psychological impacts of UPFs—such as inflammation, insulin resistance, or emotional dysregulation—may manifest differently across age groups, with younger populations being more metabolically responsive to dietary changes [23]. Recent meta-analyses have also reported that the associations between UPFs and depression, as well as metabolic diseases, are more pronounced among younger adults [14,19]. Thus, age-related dietary disparities warrant specific investigation when examining UPF as a modifier in the bidirectional relationship between depression and diabetes.

Brazil, as a large and diverse middle-income country, is experiencing a simultaneous increase in the burden of depression and diabetes. Since the 1990s, the intake of UPFs in Brazil has steadily increased, contributing from 9% to more than 20% of total energy intake currently [24,25]. Similar dietary trends and health consequences have been reported globally. In the United States, ultra-processed foods account for over 50% of total caloric intake, and their consumption has been linked to increased risk of both depressive symptoms and type 2 diabetes [17,19]. In South Korea and Italy, national surveys have also shown significant associations between UPF consumption and mental health outcomes [15,16]. A recent meta-analysis encompassing multiple countries found that higher UPF intake is associated with a 12–30% increased risk of depression and metabolic diseases, particularly in younger adults [14,19]. Given this rise in UPF consumption alongside the increasing prevalence of these health conditions, and the fact that evidence from low- and middle-income countries remains scarce, it is important to examine how UPF-rich dietary patterns may influence the relationship between depression and diabetes in the Brazilian population. This study analyzed nationally representative data to investigate whether ultra-processed food intake modifies the bidirectional relationship between depression and diabetes in Brazilian adults. This study addresses the limitations of previous research on the relationship between depression and diabetes in Brazil, such as small sample sizes and limited data on dietary factors [26]. By focusing on the role of ultra-processed food consumption as a potential modifier in this relationship, our findings may contribute to a better understanding of how dietary patterns impact depression and diabetes in developing countries like Brazil.

## 2. Materials and Methods

### 2.1. Study Sample

This study used data from the National Health Survey (Pesquisa Nacional de Saúde, PNS) conducted in 2019. PNS collected data on anthropometric measures, non-communicable diseases, and lifestyle behaviors (such as physical activity, smoking, alcohol use, and food consumption). In collaboration with the Ministry of Health, the Brazilian Institute of Geography and Statistics (Instituto Brasileiro de Geografia e Estatística, IBGE) orchestrated the development of this survey to ensure a representative sample of the Brazilian population aged 15 years and above. The data used in this study is de-identified and is available in the public domain. Further details about the National Health Survey were reported elsewhere [27].

The PNS provides weights for each participant’s measurement subsample, which are calculated based on the probability of selection of these units for the main and study samples. This is due to the fact that the PNS uses stratified sampling: i.e., a number of households are selected in a given area, and a particular member of the household is selected. The weights of the measurement subsamples therefore reflect the weight of that participant in the household in which he or she is located, the probability of selecting that household within that district, adjusted for the gender and age group (15 to 17, 18 to 24, 25 to 39, 40 to 59, and 60 or over) of the participant. We took weight into account in the statistical calculations by using Stata 18.0’s svyset command to adequately reflect the representativeness of each participant.

The investigation focused on participants who met the eligibility criteria of being at least 18 years old and having complete data on NCDs and UPF intake, resulting in a final analytic sample of 81,524 individuals out of the total 90,846 participants. For the adjusted regression models, additional requirements included the availability of anthropometric data (to ensure BMI could be calculated), as well as food intake and lifestyle information, yielding a subsample of 62,187 participants. The detailed selection process for eligible participants is illustrated in Figure 1.

### 2.2. Measures

#### 2.2.1. Independent and Dependent Variables: Depression and Diabetes

Considering the defined association between diabetes and UPF, we included depression as an independent variable and diabetes as a dependent variable in this study. The presence of diseases was self-reported and evaluated through the survey question “Have you ever been informed by a medical professional that you have depression/diabetes?”. In the case of diabetes, women who indicated that they had solely received a diagnosis during their pregnancy were excluded from the analysis.

#### 2.2.2. Effect Modifier: Frequency of UPF Intake

Considering the health effects of different types of food, we controlled the intake of plant-based and animal-based whole food while analyzing UPF. Participants were asked about the frequency of food intake using the validated qualitative Food Frequency Questionnaire (FFQ) [27], in which a number from 0 to 7 indicated the number of days in a week that these specific food groups were usually consumed: beans, vegetables, fruits, natural fruit juice, artificial fruit juice, fish, red meat, chicken, soda, milk, sweets (cakes or pies, chocolates, candies, cookies or sweet biscuits), and substitutions for meals (pre-packaged or processed food products that are intended to replace traditional meals. These often include meal replacement shakes, bars, or other convenience foods that are marketed as complete meals). We divided the foods mentioned in the questionnaire into three categories: plant-based whole foods (4 items: beans, vegetables, fruits, and natural fruit juice), animal-based whole foods (4 items: fish, milk, red meat, and chicken), and UPF (4 items: soda [except diet soda], artificial fruit juice [except diet juice], sweets, and substitutive meals) [13]. For plant-based and animal-based whole foods, we set low/high intake thresholds for each food group according to optimal intake values carried out by Lukas et al. [20], then combined each intake result to classify the study sample into low, medium, and high consumption. For UPF, thresholds for low/high intake of each food were defined by the median of the study sample, and the study sample was divided into low and high consumption according to two classification methods—UPF1 and UPF2—based on different thresholds of weekly consumption, for high/low UPF intake. Specifically, UPF1 defines high consumption as any UPF having a weekly consumption frequency higher than the median value for the population, while UPF2 defines high consumption as all UPFs having weekly consumption frequencies higher than the median value for the population. Appendix A records the detailed low/high intake thresholds and the classification methods of each food group. In this study, UPF consumption category was a potential effect modifier, while plant-based and animal-based consumption categories were covariates. To address concerns about multicollinearity of the combined variables, we calculated the variance inflation factor (VIF) between the original FFQ variables and recorded the mean values in Appendix A as well.

#### 2.2.3. Covariates

In addition to plant-based and animal-based consumption categories, the covariates in our study include demographic and socioeconomic variables such as age, sex, geographical region (North, Northeast, South, Southeast, Midwest), type of residence (urban or rural), race (white or non-white), marital status, household income per capita (<2, ≥2 but <3, ≥3 minimum wages of Brazil (BRL 998, USD 254 by average exchange rate, in 2019) [21], and the highest education level (elementary school or less, high school, university, or more).

We also included the health-related variables such as smoking status (ever or never smoked) and alcohol consumption (yes or no in the last month), and, based on the survey questions, we introduced obesity and physical activity covariates: from the self-reported height and weight of the participants body mass index (BMI, kg/m^2^) was calculated. Obesity was defined as a BMI greater than or equal to 30 kg/m^2^; [22] by calculating the participants’ weekly time spent on leisure exercise (such as going to the gym), heavy lifting activities, and commuting by walking or cycling, we defined participants as physically active if their total time spent on the above activities per week was greater than or equal to 150 min [23].

### 2.3. Statistical Methods

Univariate descriptive statistics were conducted to calculate means, standard deviations (SD) of age, and the percentage of population with different characteristics in the sample. Bivariate descriptive statistics were conducted to compare average age of people with and without diabetes, and prevalence of diabetes between participants with different characteristics. We used ANOVA tests for the average age of diabetes groups, and chi-square tests for other categorical variables. Demographic and socioeconomic variables that showed significant difference in prevalence of diabetes between groups in the bivariate descriptive analyses (*p* < 0.05) were adjusted for in further regression analyses.

Multiple logistic regression analyses having diabetes as the dependent variable were performed. The preliminary multiple logistic regression analysis involved 10 models, 5 models for each of the 2 UPF consumption classification methods as follows: (1) depression and UPF consumption categories (unadjusted); (2) depression, UPF consumption categories, and selected covariates (basic); (3) Model 2 with multiplicative interaction term for depression and UPF consumption categories (interacted); (4) depression and selected covariates, only including low UPF consumption population (stratified—low consumption); (5) depression and selected covariates, only including high UPF consumption population (stratified—high consumption). Considering that age is closely related to depression and diabetes [9], for further stratified analysis, the population was divided into old adults (≥60 years old) and non-old adults (<60 years old) according to the Brazilian standards [28].

The preliminary analysis revealed key patterns that shaped our final model selection. First, the interaction between depression and UPF consumption categories indicated that the association between depression and diabetes might be stronger in individuals with higher UPF intake. This justified exploring the role of UPF consumption as a potential modifier. Additionally, when stratifying by UPF consumption, we observed a stronger association between depression and diabetes in the high UPF group, suggesting that higher UPF intake could amplify diabetes risk among those with depression. Lastly, age stratification showed that the relationship between depression, UPF intake, and diabetes was more pronounced in the old adult population, highlighting the importance of considering age as a factor. These findings led us to refine our models to better capture these nuanced interactions. Therefore, further stratified analysis involved 8 logistic regression models (2 UPF consumption classification methods × 2 age categories × 2 UPF consumption categories), which were models with depression and adjusted for selected covariates. We tested a three-way interaction between depression, UPF intake, and age in the fully adjusted model. The interaction term was statistically significant (*p* = 0.037), indicating that the modifying effect of UPF intake on the depression–diabetes association differs across age groups. Based on this result, we proceeded with stratified analyses by age to explore these patterns in more detail.

All analyses were conducted using Stata 18.0 for Mac (Stata Corp, College Station, TX, USA), and statistical tests were two-sided at the *p* < 0.05 level. In logistic regression models, the effect size is expressed as the minimum detectable odds ratio per one SD increase in the continuous explanatory variable [29]. The minimum detectable odds ratio to check the effect size of our study is 1.0355.

### 2.4. Ethical Approval

The study used publicly available, de-identified secondary data from the Brazilian National Health Survey (PNS 2019). Ethical approval for public use of these data has been granted by the Brazilian Institute of Geography and Statistics (IBGE) and the Brazilian Ministry of Health. No further ethical approval was required for this secondary data analysis.

## 3. Results

### 3.1. Characteristics of Participants

The characteristics of the sample are shown in Table 1. Univariate descriptive statistics showed that the overall prevalence of diabetes in the sample was 8.68%, the prevalence of depression was 9.82%, and the prevalence of obesity was 21.11%. In the two UPF consumption classification methods, the populations defined as high consumption were 69.33% (UPF1) and 4.25% (UPF2) of the total sample. Physically active participants comprised 33.42% of the respondents based on self-reported activity ≥150 min per week, 78.25% of the sample’s household income per capital was less than BRL 1996 per month (USD 507 by average exchange rate in 2019), most of the respondents live in cities, and the average age of the sample was 47 years old, ranging from 18 to 92 years. Bivariate descriptive statistics showed that people who were female, with lower levels of education, lived in the Southeastern region and urban area, and were married had a higher prevalence of diabetes. For health-related status and behaviors, individuals with a higher prevalence of diabetes in our sample were more likely to report depression, obesity, and low UPF consumption, as well as higher intake of plant-based whole foods. They were also more likely to have smoked, be physically inactive, and not have consumed alcohol in the past month. The average age of people with diabetes was higher than that of people without diabetes.

We then confirmed the covariates other than household income per capita and race in Table 2 as selected covariates, based on the *p*-value cutoff point at 0.05, and included them in the adjusted regression models, based on the statistical significance.

### 3.2. Effect of UPF Intake on Comorbidities

Table 2 (see Appendix A for details) illustrates the results of the preliminary logistic regression models for diabetes. When we use UPF1 to define the high and low consumption of UPF, the unadjusted model showed that participants with depression have a higher likelihood of having diabetes (OR: 1.515, 95% CI: 1.408 to 1.629), while those with higher UPF intake were less likely to have diabetes (OR: 0.370, 95% CI: 0.352 to 0.389), this fact is consistent when using the UPF2 for definition (OR: 1.529, 95% CI: 1.422 to 1.643 with depression, and OR: 0.371, 95% CI: 0.309 to 0.446 with UPF). This inverse association in the unadjusted model likely reflects reverse causality, whereby individuals previously diagnosed with diabetes may have reduced their UPF intake as part of dietary management recommendations. After including selected covariates, participants with depression remained significantly more likely to have diabetes (OR: 1.258, 95% CI: 1.064 to 1.489 for UPF1, and OR: 1.251, 95% CI: 1.059 to 1.478 for UPF2), the association between high UPF consumption and a lower likelihood of diabetes remained significant under the UPF1 definition (OR: 0.547, 95% CI: 0.495 to 0.605), while this association was no longer significant under the UPF2 definition (OR: 0.796, 95% CI: 0.571 to 1.109).

The interaction terms under models with both UPF definitions were significantly and positively associated with the likelihood of having diabetes (OR: 1.379, 95% CI: 1.009 to 1.885 for UPF1, and OR: 2.900, 95% CI: 1.197 to 7.028 for UPF2). When defined by UPF1, the association between depression and diabetes in the interactive model was no longer significant (OR: 1.035, 95% CI: 0.836 to 1.281), while the strength and significance of this association in interactive model was slightly reduced when defined by UPF2 (OR: 1.209, 95% CI: 1.021 to 1.432), which is expected due to the inclusion of interaction terms.

Stratified analysis under the definition of UPF1 showed that a significant positive association between diabetes and depression was present only in those with high consumption of UPF (OR: 1.401, 95% CI: 1.107 to 1.774 for high UPF consumption vs. OR: 1.045, 95% CI: 0.844 to 1.294 for low UPF consumption). When the definition was shifted to UPF2, we found positive and significant associations between diabetes and depression in both groups after stratification, but the strength and significance of the association was higher in the high UPF consumption group (OR: 3.551, 95% CI: 1.394 to 9.046 for high UPF consumption vs. OR: 1.211, 95% CI: 1.023 to 1.434 for low UPF consumption).

### 3.3. Further Stratified Analysis Based on Age

Table 3 shows the prevalence of diabetes and depression according to the stratification of age. We found that the prevalence of depression and diabetes in old adults was higher than that in non-old adults and that people with one disease had a higher prevalence of the other disease, whether depression or diabetes. A total of 22.84% of the old adults with depression had diabetes, and 12.54% of the old adults with diabetes had depression.

Table 4 (see Appendix A for details) illustrates the results of the further logistic regression models for diabetes after being stratified by age. We found that, no matter under the definition of UPF1 or UPF2, a significant positive association between diabetes and depression was present only in the non-old adult population with high consumption of UPF (OR: 1.596, 95% CI: 1.127 to 2.260 for UPF1 and OR: 6.726, 95% CI: 2.625 to 17.233 for UPF2). In addition, when the UPF classification method and UPF consumption category were the same, the model based on the non-old adult population had a higher adjusted R-squared value than the model based on the old adult population, which means that the model had a stronger explanatory power.

## 4. Discussion

### 4.1. Summary of Findings

Employing two classification methods of UPF consumption, our study explored the effect modification influence of high UPF intake on the association between depression and diabetes. Specifically, using a broader UPF definition (i.e., defined by UPF1), the association between depression and diabetes was observed only among individuals with high UPF intake. Under the UPF2 definition, high UPF consumers had a significantly stronger depression–diabetes association than low UPF consumers.

### 4.2. Potential Mechanisms and Pathophysiological Links

As a cross-sectional study, the underlying mechanisms of this phenomenon can be discussed as a start from both depression and diabetes. From depression to diabetes, studies have suggested that stress and emotional eating in depression can increase the risk of diabetes. These eating behaviors are usually intense, uncontrolled, and tend to be associated with a higher consumption of UPFs [14]. Konttinen et al. [30] found that emotional eating was a complete mediating factor explaining the increased level of sweets consumption in patients with depression. A longitudinal study in South Korea also reported a positive association between depressive symptoms and unhealthy eating patterns, including UPF intake, among middle-aged adults [15]. In addition, use of antidepressants, including selective serotonin reuptake inhibitors (SSRIs), serotonin and norepinephrine reuptake inhibitors (SNRIs), tricyclic antidepressants, and sodium salts, especially when used in the long-term or at higher doses, has also been associated with new-onset diabetes [31]. A U.S. study found that for adolescents who used an SSRI or SNRI for more than 150 days, there was a 2.39-fold increased risk of developing type 2 diabetes with medication more than 15 mg/day (in fluoxetine hydrochloride dose equivalents) compared with those who had 15 mg/day or less, though this association may depend on duration and dosage, and is not universally observed [32].

From diabetes to depression, ultra-processed foods have become increasingly common in the diets of young people due to industrialized food systems and lifestyle shifts [33]. Diabetes may also increase the risk of depression. This bidirectional association has been confirmed in studies from both high-income and middle-income countries. For instance, a large European study (DEPLAN) reported that people with early-onset diabetes exhibited a higher rate of symptoms related to depression [34]. Studies have shown that diabetes diagnoses at a young age are associated with higher levels of depressive symptoms because these people typically suffer from greater levels of internalized stigma and self-blame [35]. This also explains the significance of the association between depression and diabetes observed only in non-old adults with high UPF consumption when we stratified based on age. Additionally, people with diabetes may experience poor glycemic control when UPF is highly consumed, leading to frustrated feelings, which may eventually develop into depression [36]. Some potential mechanisms have also been proposed in terms of pathophysiology, such as systemic inflammatory responses mediated by immune factors such as C-reactive protein, tumor necrosis factor-α (TNF-α), interleukin 1 (IL-1), and IL-6. Ultra-processed foods may contribute to this inflammatory response by promoting gut dysbiosis, increasing intestinal permeability (“leaky gut”), and elevating circulating endotoxins and pro-inflammatory cytokines [29]. In addition, chronic psychological stress and poor dietary quality can activate the hypothalamic-pituitary-adrenal (HPA) axis and the sympathetic nervous system (SNS), leading to dysregulated cortisol production and insulin resistance—both of which are implicated in the pathogenesis of depression and diabetes. These shared mechanisms support the plausibility that UPF consumption exacerbates biological vulnerabilities for both conditions, especially in younger individuals who may be more susceptible to neuroendocrine and metabolic disruption [9,37].

### 4.3. Age-Stratified Differences

Stratification of results according to age revealed that the effects we found were only present in the non-old adult population fed with a high UPF diet. This is consistent with previous findings showing that diabetes is a disease highly associated with older age [9]. The etiology of diabetes in older populations may be more complex than in non-old adult populations. Old adults are a special subgroup of patients with diabetes. Current studies have shown that with advancing age, there are many physiological mechanisms, such as insufficient insulin secretion, as well as increased insulin resistance caused by changes in body composition and sarcopenia, leading to a higher prevalence of diabetes in the old adults [38]. The complexity of diabetes in old adults also supports the fact that the same model in our study has lower explanatory power for old adults than for the non-old adult population, indicating the need for clinicians to have age-specific discussions in the patient care of diabetes.

It is important to emphasize that people who have neither diabetes nor depression should also reduce their intake of UPF, because the relationship between UPF and various chronic non-communicable diseases has been increasingly defined. Its unbalanced nutritional composition, chemical modification, use of special additives, and potential exposure to endocrine disruptors resulting from the intake of plastic ingredients from excessive packaging, such as phthalates or bisphenol A, may be important risk factors for metabolic disease development [19]. Therefore, the public should reduce the intake of related foods, and regulatory agencies should formulate policies and regulations to guide the public to choose low-processed foods. We also want to draw attention to the complexity of the causes of diabetes across the whole population, even among non-old adult groups, with identified risk factors encompassing complex genetic-environmental interactions and others such as obesity and sedentary lifestyles [39]. We demonstrated differences in the effect of ultra-processed foods on depression–diabetes comorbidity relationship between different populations, which is not inconsistent with existing evidence-based causes and perspectives.

### 4.4. Contributions and Implications for Future Research

This study gives health professionals the opportunity to understand the role of UPF as an effect modifier in the relationship between depression and diabetes and suggests that the importance of UPF intake should be considered in the care of individuals at risk for diabetes, comorbidity prevention and diagnosis, for both patients with depression and diabetes. In addition, we also emphasize emotional eating interventions and concerns about drug concomitant usage for patients with depression, and psychological interventions for patients with diabetes, especially younger patients.

This study shed light on new insights in the relationship between depression and diabetes and found an effect-modifying role of UPF intake. This is different from previous studies as it provides new insights and inspiration for subsequent pathway studies (such as moderator and mediator analysis). Second, while using PNS 2019, the Brazilian national database, we provided developing country and global South perspectives for metabolic disease, and depression–diabetes comorbidities studies with a large and representative data source. Finally, in terms of practicality, we propose the indication to develop new studies in the subject to define how to formulate new guidance and recommendations for clinicians, which need to be further validated.

### 4.5. Limitations

Limited by the variables of FFQ affiliated with the PNS, the UPFs identified in this study may not address all aspects of dietary patterns of the population. Another limitation was the potential bias that may exist in the relationship between UPF consumption and diabetes. Our results showed that individuals with high UPF consumption had a lower prevalence of diabetes, which could be attributed to the cross-sectional nature of our data. This may reflect reverse causality, where individuals diagnosed with diabetes adopt healthier diets, leading to lower reported UPF consumption. This epidemiological fallacy has also been raised in other studies involving diabetes. Furthermore, the utilization of patients’ self-reported data concerning dietary habits and diagnosed illnesses may introduce recall bias. The evaluations of dietary intake were constrained, lacking a comprehensive representation of the full spectrum of foods ingested by the participants, such as portion size, nutrient density, and calorie information. Additionally, the PNS 2019 dataset solely encompassed data on the frequency of food consumption, omitting crucial particular aspects like body weight and calorie consumption. To enhance our comprehension of the noted correlation and the plausible enduring repercussions of diverse dietary patterns on health outcomes, it is imperative to conduct additional investigations that delve into the underlying mechanisms.

## 5. Conclusions

Our findings demonstrate that ultra-processed food (UPF) intake significantly modifies the relationship between depression and diabetes in a large Brazilian adult population. Age-stratified analyses revealed that this effect modification was particularly pronounced in younger adult groups. These results suggest that dietary assessment and nutrition-focused interventions—particularly targeting UPF consumption—should be integrated into clinical guidelines for the management and prevention of comorbid depression and diabetes, particularly for younger populations. Health professionals should consider routine screening for dietary patterns in patients presenting with either condition and deliver tailored counseling to reduce UPF intake, especially among younger adults at heightened risk. From a public health and policy perspective, our findings reinforce the need for multi-level strategies to address high UPF consumption. These include public education campaigns to raise awareness, clear front-of-package labeling, restrictions on marketing UPFs to children and vulnerable populations, and taxation policies that incentivize healthier food choices. Incorporating mental and metabolic health considerations into national dietary guidelines may also help address the rising burden of non-communicable diseases. Given the increasing global burden of non-communicable diseases and the expansion of UPFs in global food systems, these efforts are especially urgent in low- and middle-income countries undergoing dietary transitions. Future longitudinal studies are needed to assess causal direction and to further explore whether UPF intake serves as a moderator or potentially a mediator in the depression–diabetes relationship. Such work will be critical in designing targeted, evidence-based interventions aimed at reducing the joint burden of mental and metabolic disorders.

## Figures and Tables

**Figure 1 nutrients-17-02454-f001:**
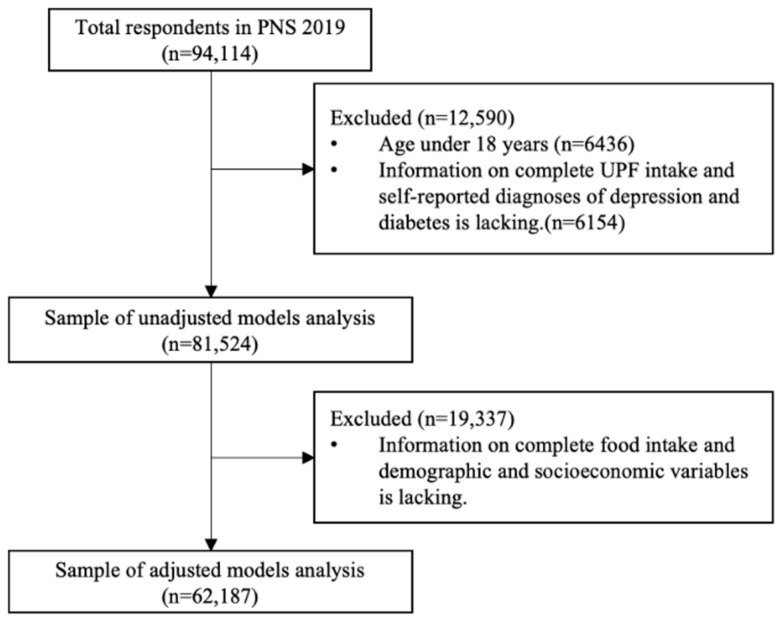
Participant flow chart.

**Table 1 nutrients-17-02454-t001:** Characteristics of Brazilian adults aged 18 years and above by diabetes: PNS, 2019.

	Total (n = 81,524)		Non-Diabetes (n = 74,448, 91.3%)		Diabetes (n = 7076, 8.7%)		*p* ^a^	*N*
	n (M) ^b^	%(SD)	n (M)	%(SD)	n (M)	%(SD)		
1. Depression							<0.001	81,524
No	73,520	90.18	67,418	91.70	6102	8.30		
Yes	8004	9.82	7030	87.83	974	12.17		
2. Dietary patterns								
UPF consumption (UPF1) **^c^**							<0.001	81,524
Low	25,002	30.67	21,317	85.26	3685	14.74		
High	56,522	69.33	53,131	94.00	3391	6.00		
UPF consumption (UPF2)							<0.001	81,524
Low	78,061	95.75	71,106	91.10	6955	8.90		
High	3463	4.25	3342	95.97	121	4.03		
Plant-based whole food consumption							<0.001	81,524
Low	6637	8.14	6151	92.68	486	7.32		
Intermediate	67,161	82.38	61,391	91.41	5770	8.59		
High	7726	9.48	6906	89.39	820	10.61		
Animal-based whole food consumption							0.17	81,524
Low	4491	5.51	4118	91.69	373	8.31		
Intermediate	74,776	91.72	68,291	91.33	6485	8.67		
High	2257	2.77	2039	90.34	218	9.66		
3. Health								
Smoking status							<0.001	71,752
Never	49,321	68.74	45,759	92.78	3562	7.22		
Ever	22,431	31.26	19,596	87.36	2835	12.64		
Alcohol consumption last month							<0.001	81,524
No	49,618	60.86	44,245	89.17	5373	10.83		
Yes	31,906	39.14	30,203	94.66	1703	5.34		
Obesity							<0.001	81,524
No	64,317	78.89	59,472	92.47	4845	7.53		
Yes	17,207	21.11	14,976	87.03	2231	12.97		
Physically active							<0.001	81,524
No	54,282	66.58	48,861	90.01	5421	9.99		
Yes	27,242	33.42	25,587	93.92	1655	6.08		
4. Demographic and socioeconomic status								
Sex							<0.001	81,524
Male	37,378	45.85	34,500	92.30	2878	7.70		
Female	44,146	54.15	39,948	90.49	4198	9.51		
Race							0.06	81,517
White	30,577	37.51	27,848	91.07	2729	8.93		
Non-white	50,940	62.49	46,593	91.47	4347	8.53		
Region							<0.001	81,524
North	14,830	18.19	13,788	92.97	1042	7.03		
Northeast	27,970	34.31	25,557	91.37	2413	8.63		
Southeast	18,579	22.79	16,751	90.16	1828	9.84		
South	10,656	13.07	9690	90.93	966	9.07		
Central	9489	11.64	8662	91.28	827	8.72		
Highest education obtained							<0.001	70,659
Elementary school-	31,618	44.75	27,794	87.91	3824	12.09		
High school	25,142	35.58	23,706	94.29	1436	5.71		
University+	13,899	19.67	13,113	94.34	786	5.66		
Household income per capita							0.14	81,504
<2 × minimum wage	63,995	78.52	58,504	91.42	5491	8.58		
2~3 × minimum wage	7215	8.85	6565	90.99	650	9.01		
≥3 × minimum wage	10,294	12.63	9359	90.92	935	9.08		
Residence							<0.001	81,524
Urban	64,080	78.60	58,350	91.06	5730	8.94		
Rural	17,444	21.40	16,098	92.28	1346	7.72		
Marital status							<0.001	81,524
Married	32,985	40.46	29,890	90.62	3095	9.38		
Other	48,539	59.54	44,558	91.80	3981	8.20		
Age	47.36	17.09	46.66	16.77	62.07	12.96	<0.001	81,524

^a^ *p*-values were derived from the results of appropriate tests comparing given variable among people with and without diabetes. We used the chi-square tests for all variables, except age, for which we used ANOVA test. ^b^ All variables are presented as count (n) and percentage (%), except for age, which is expressed as mean (M) and standard deviation (SD) due to its continuous nature. ^c^ UPF1 refers to a flexible definition of high ultra-processed food intake, based on frequent consumption of at least one UPF item, while UPF2 represents a stricter definition requiring frequent intake of all UPF categories to reflect a consistently high-UPF dietary pattern.

**Table 2 nutrients-17-02454-t002:** Odds ratios for diabetes related to depression among different UPF consumption populations.

Model	No Depression	Odds Ratio (95% CI)	*p*
		Depression	
UPF1			
Unadjusted	Reference	1.515 (1.408, 1.629)	<0.001
Basic	Reference	1.258 (1.064, 1.489)	0.01
Interactive	Reference	1.035 (0.836, 1.281)	0.71
Stratified—low consumption	Reference	1.045 (0.844, 1.294)	0.66
Stratified—high consumption	Reference	1.401 (1.107, 1.774)	<0.01
UPF2			
Unadjusted	Reference	1.529 (1.422, 1.643)	<0.001
Basic	Reference	1.251 (1.059, 1.478)	0.01
Interactive	Reference	1.209 (1.021, 1.432)	0.02
Stratified—low consumption	Reference	1.211 (1.023, 1.434)	0.02
Stratified—high consumption	Reference	3.551 (1.394, 9.046)	0.01

**Table 3 nutrients-17-02454-t003:** Diabetes and depression prevalence details stratified by age. (*N* = 81,524).

	Depression −	Depression +	Total
Age 18–59			
Total	53,842	5657	59,499
Diabetes+	2364	438	2802
% of diabetes	4.39	7.74	4.71
Age ≥ 60			
Total	19,678	2347	22,025
Diabetes+	3738	536	4274
% of diabetes	19.00	22.84	19.41

**Table 4 nutrients-17-02454-t004:** Odds ratios for diabetes related to depression among different UPF consumption populations, stratified by age.

Models	No Depression	Odds Ratio (95% CI)	p	R^2^ (%, Adjusted)
		Depression		
UPF1				
low consumption, 18–59 years	Reference	0.941 (0.682, 1.298)	0.61	12.83
low consumption, 60+ years	Reference	1.075 (0.813, 1.420)	0.82	3.36
high consumption, 18–59 years	Reference	1.596 (1.127, 2.260)	0.01	12.49
high consumption, 60+ years	Reference	1.112 (0.840, 1.472)	0.40	2.73
UPF2				
low consumption, 18–59 years	Reference	1.238 (0.947, 1.618)	0.11	13.51
low consumption, 60+ years	Reference	1.098 (0.900, 1.339)	0.32	2.37
high consumption, 18–59 years	Reference	6.726 (2.625, 17.233)	<0.001	14.37
high consumption, 60+ years	Reference	0.323 (0.036, 2.865)	0.31	12.53

## Data Availability

The data analyzed in this study are publicly available from the Brazilian National Health Survey (Pesquisa Nacional de Saúde—PNS 2019), accessible through the Brazilian Institute of Geography and Statistics (IBGE) official website at https://www.ibge.gov.br. Researchers can obtain these datasets upon request following IBGE’s data access guidelines and conditions. The code used for data analysis is available from the corresponding author upon reasonable request.

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
