# Peer review of "Ultra-Processed Food Intake as an Effect Modifier in the Association Between Depression and Diabetes in Brazil: A Cross-Sectional Study"

_nutrients, 2025, doi:10.3390/nu17152454_

Round 1

Reviewer 1 Report

Comments and Suggestions for Authors

In my point of view, the work conducted by Sun and collaborators can be considered for publication in Nutrients, after the authors perform the following revisions:

The abstract is well-written and gives a good illustration of the developed work. I only suggest the inclusion of some directions for further studies at the end of the conclusions.

In the introductory section you should indicate the most relevant data on other regions worldwide and not only to focus in Brazil.

In the Materials and Methos section you have to provide the ethical details.

The Results are fine and properly described, but the tables have to be properly formatted.

In the Discussion, I suggest its division into subsections, aligned with the Results. Once again, more data from other studies conducted worldwide should be discussed. This section can be improved.

A section on study’s strengths and limitations shoul be included.

At the end of your Conclusions, future perspectives are missing. You can also add some practical implications.

Author Response

Comment 1: The abstract is well-written and gives a good illustration of the developed work. I only suggest the inclusion of some directions for further studies at the end of the conclusions.

Response 1: We added the following direction at the end of the revised abstract: “Future longitudinal studies are warranted to establish causality, investigate underlying biological mechanisms, and examine whether improving overall nutrient intake through dietary interventions can reduce the co-occurrence of depression and diabetes.” (line 43-45).

Comment 2: In the introductory section you should indicate the most relevant data on other regions worldwide and not only to focus in Brazil.

Response 2: In response, we have revised the final paragraph of the Introduction to incorporate global data on ultra-processed food consumption and its association with depression and diabetes. We now reference findings from the United States, South Korea, and Italy, as well as a recent meta-analysis covering multiple countries, to provide a broader international context (line 105). The addition of this data aim to situate our study within the global evidence base and highlight the relevance of our findings beyond Brazil.

Comment 3: In the Materials and Methos section you have to provide the ethical details.

Response 3: We have added 2.4. Ethnical Approval (line 255) at the end of the Materials and Methods section.

Comment 4: The Results are fine and properly described, but the tables have to be properly formatted.

Response 4: In response to your suggestion, we have revised the formatting of all tables to align with the journal’s guidelines. Specifically, in Table 1, we removed informal horizontal lines within the body of the table, consolidated the presentation of count and percentage [n (%)] as well as mean and standard deviation [M (SD)] where appropriate, and added a clarifying footnote to enhance readability and consistency. (line 285)

Comment 5: In the Discussion, I suggest its division into subsections, aligned with the Results. Once again, more data from other studies conducted worldwide should be discussed. This section can be improved.

Response 5: We have divided the results section into different subsections. We also cited and discussed more studies, such as Lee and Choi [15] (line 369) and Nouwen et al. [35] in 4.2. (line 383)

Comment 6: A section on study’s strengths and limitations shoul be included.

Response 6: We now added (page13, Line 434-471).

Comment 7: At the end of your Conclusions, future perspectives are missing. You can also add some practical implications.

Response 7: We included both future research perspectives and practical implications, as well as policy perspective. We also highlighted the need for longitudinal studies to clarify causal pathways and determine whether UPF intake serves as a mediator or moderator in the relationship between depression and diabetes (line 491-495).

Reviewer 2 Report

Comments and Suggestions for Authors

This manuscript investigates the role of ultra-processed food (UPF) intake as a potential effect modifier in the relationship between depression and diabetes in a large, nationally representative sample of Brazilian adults. The topic is timely, original, and of public health relevance. The study uses a robust dataset and appropriate statistical methods, including interaction terms and stratified logistic regression. 

The manuscript requires substantial language editing to improve clarity, grammar, and academic tone. Many sentences are overly long, awkwardly phrased, or redundant (e.g., in the introduction and abstract). I strongly recommend professional English-language editing prior to resubmission.

The introduction could be more concise by reducing repeated definitions of UPFs and combining some epidemiological context more effectively.

Strengthen the conclusion by more clearly articulating what public health or clinical interventions could be informed by these findings.

The paper has strong potential but requires minor revision in clarity, interpretation, and presentation to reach publishable quality.

Author Response

Comment 1: The manuscript requires substantial language editing to improve clarity, grammar, and academic tone. Many sentences are overly long, awkwardly phrased, or redundant (e.g., in the introduction and abstract). I strongly recommend professional English-language editing prior to resubmission.

Response 1: Thank you for pointing out our language issues. We consulted native English speakers to revise the language, particularly in the introduction and abstract sections and added changes to the manuscript.

Comment 2: The introduction could be more concise by reducing repeated definitions of UPFs and combining some epidemiological context more effectively.

Response 2: We have revised the Introduction to improve conciseness by reducing repetitive definitions of ultra-processed foods (UPFs) and streamlining the epidemiological context. Specifically, we merged overlapping information about UPF characteristics and prevalence, and we clarified the global and national trends in depression, diabetes, and UPF consumption in a more integrated and efficient manner (line 73-85).

Comment 3: Strengthen the conclusion by more clearly articulating what public health or clinical interventions could be informed by these findings.

Response 3: In response, we made more clearly the directions of definition of public health recommendations and potential interventions, strengthening the conclusion by more clearly articulating the public health and clinical implications of our findings. Specifically, we now recommend that dietary assessment and counselling on UPF consumption be incorporated into routine clinical care for individuals at risk of depression-diabetes comorbidity, particularly among younger adults. From a public health perspective, we suggest actionable strategies such as public education campaigns, front-of-package food labelling, marketing restrictions, and fiscal policies to reduce UPF intake. (line 484-486) These proposed interventions align with the growing need for multi-level approaches to address the rising burden of mental and metabolic disorders in developing countries.

Reviewer 3 Report

Comments and Suggestions for Authors

Overall Comments

This is a timely and important study that addresses a critical and understudied public health question in the context of Brazil: the intersection between ultra-processed food (UPF) consumption, depression, and diabetes. The use of a nationally representative dataset (PNS 2019) and the stratified analyses by age add strength and relevance. The paper has the potential for high impact, especially in the fields of global health and nutrition epidemiology. However, the methods section lacks important information that precludes readers to assess rigor of the study and potential for bias. For instance, more information on how the variables were operationalized, age range, and rationale for the selection of the statistical models employed would enhance the paper. The introduction and the methods lack rationale for the stratified analysis for age. I wonder if the age-stratified analysis was not part of the original plan for this paper. Perhaps not including the age-stratified analysis/results would improve the focus of the paper in terms of results and discussion on the effect modification by UPF only.

  1. Abstract Review

Strengths:

  • Appropriately situates the study in a large, representative dataset. Including demographic and dietary variables is a strength.
  • Summarizes the key results clearly with odds ratios and 95% CIs, including age-stratified findings.
  • The conclusion offers appropriate implications, calling for future longitudinal studies.

Comments & Recommendations

  • Lines 17-18: The background line does support the need for the study if this association has been previously investigated.
    • Recommendation: Highlight the gap in the literature and why study disparities in age.
  • Line 22: Given age is an important variable in this study, I suggest authors to include the age range of the adult population in the abstract.
  • Methods: Suggest including how diet was assessed (qualitative FFQ).
  • Lines 25–26: “employing two classification methods for UPF intake (UPF1 and UPF2)” — needs a little explanation.
    • Recommendation: Add a clause to clarify, e.g., “based on different thresholds of weekly consumption.”
    • Unclear what 1 and 2 stands for, as in Line 27 authors use the term “high UPF”. Is it high in both UPF 1 and 2 or does this refer to something different? Recommend consistency among the terms and a brief explanation.

  • Lines 28–29: “Adjusted models demonstrated elevated odds ratios (OR)...” —  It was a little unclear which reference group was being used and which variables were adjusted for.
    • Recommendation: Specify the comparison (e.g., “compared to individuals with low UPF intake”) and clarify the model (e.g., adjusted for age, sex, etc.).

  • Line 34–36: “These results emphasize the critical role dietary patterns may play in mental and metabolic health…”
    • Recommendation: Strengthen the takeaway: “These findings suggest that high UPF intake may influence the relationship between depression and diabetes, especially in younger adults.”

  1. Introduction Review

Strengths:

  • Strong presentation of the burden of depression globally and in Brazil.
  • Appropriately highlights the importance of comorbidity between depression and diabetes.
  • Good foundational information on UPF characteristics and their known associations with chronic diseases.
  • Clearly states the rationale for examining this relationship in the Brazilian context, using nationally representative data.

Comments & Recommendations

  • The introduction is lacking literature on age-related disparities in terms of diet, and why it could be a variable to examine in the association between depression and diabetes modified by UPF.
  • Line 43: “Depression epidemic deserves more attention in the post-pandemic era...”
    • Recommendation: Rephrase: “The global burden of depression has grown significantly in the post-pandemic era...”

  • Line 66: “UPF originally refer to foods containing a variety of unnatural additives.”
    • Comment: I suggest using the most recent definition here. Include a citation for the reference. For example: “Ultra-processed foods (UPFs), as defined by the NOVA classification, are industrial formulations typically containing additives, colorants, and emulsifiers not commonly found in home cooking.” Please include citations in all sentences from line 66-72.
  • Line 73-74: Reconsider the use of “While” in the beginning of the sentence, it does not read well. Perhaps shorten and combine the sentence with the previous one.

  • Lines 84–85: “Since the 1990s, the intake of UPFs in Brazil has steadily increased.”
    • Recommendation: Add a supporting statistic or trend line if possible (e.g., percentage increase in caloric contribution).

  • Lines 86–89: “Using Brazilian nationally representative data, we investigated the association between depression and diabetes in Brazil and how ultra-processed food diet affect this relationship.”
    • Recommendation: Revise: “This study uses nationally representative data to investigate whether ultra-processed food intake modifies the relationship between depression and diabetes in Brazilian adults.”

  • Methods Section Review

Strengths:

  • Use of a nationally representative dataset (PNS 2019) is a major methodological strength.
  • The study sample is large, well-described, and includes stratification weights.
  • Stratified regression models and interaction terms are appropriate for the research question.
  • Adjustment for a wide range of demographic, behavioral, and health covariates increases validity.

Comments & Recommendations

Study Sample

  • Line 98–99: The phrase “participants’ responses, about their anthropometrics, noncommunicable diseases, and lifestyle habits...”.
      • Recommendation: Rephrase to: “This survey collected data on anthropometric measures, noncommunicable diseases, and lifestyle behaviors.”
  • The authors mention here and in the abstract that participants with complete anthropometric data were retained. However, it is unclear why anthropometric was used as the eligibility criteria. Furthermore, this information is not included in Figure 1. Please include the rationale for selecting participants with complete anthropometrics given this is a covariate.
  • Figure 1: suggest saying “missing” rather than “lacking”. Why use the UPF variable in one box and food intake in the second excluded box?

Weighting

  • Line 114: Typo in “We taken weight into account…”
      • Recommendation: “We took weights into account…”
  • Comment: Consider explaining how the weights were incorporated (e.g., “weights were applied using Stata’s svyset command”).

Measures

  • Line 132: I believe there is a copy/paste error – do authors mean ‘Effect modifier: Frequency of UPF intake’?

Food Group Classification

  • Line 135: please include number of food items, whether this was a qualitative or semi-quantitative FFQ (it seems like it is a qualitative), and the timeframe for responses (past week?).
  • Lines 142–145: UPF classification includes soda and artificial juice, which is common in NOVA. Consider explicitly citing NOVA here for transparency.
  • Include the definitions of UPF1 and UPF2.

Covariates

  • What is the oldest age of the participants in this study? From Table 1 it seems like about 75 y/o. I ask here because I wonder if authors considered a different obesity cut-off for older adults (> 65 y/o)?
    • Example/reference Esparza-Hurtado, N., Martagon, A.J., Hart-Vazquez, D.P. et al.Novel BMI cutoff points for obesity diagnosis in older Hispanic adults. Sci Rep 14, 27498 (2024). https://doi.org/10.1038/s41598-024-65553-

Statistical Methods

  • Line 183: Please include the rationale for using logistic regression models that estimate odds ratio rather than a model to estimate prevalence ratio (e.g. log-binomial models).
    • Citation: Barros, A.J., Hirakata, V.N. Alternatives for logistic regression in cross-sectional studies: an empirical comparison of models that directly estimate the prevalence ratio. BMC Med Res Methodol3, 21 (2003). https://doi.org/10.1186/1471-2288-3-21
  • Line 191: Did authors explore a three-way interaction term between depression*UPF*age before stratifying by age? Please describe and include the p for this 3-way interaction to help support the stratified analysis.

  1. Results Section Review

Strengths:

  • Large, representative sample size is maintained throughout analyses (n = 81,524).
  • Results are clearly divided into subsections: characteristics, regression analyses, and age-stratified models.
  • Age-stratified analysis is a particularly valuable addition for public health relevance.

Comments & Recommendations

Participant Characteristics

  • Please include range of age of participants
  • Table 1 is not stand-alone as is. Please include: n and % of diabetes/non-diabetes, definition of UPF 1 and 2, describe abbreviations as footnote
  • Line 218: “Physically active participants comprised 33.42%...”
      • Add a clarifying statement, e.g., “...based on self-reported activity ≥150 minutes per week.”

  • Lines 224–227: The association findings (e.g., “higher prevalence of diabetes among those physically inactive, obese, or with depression...”) are meaningful but buried in a long sentence.
      • Recommendation: Consider separating these associations by variable group (e.g., behavioral vs. sociodemographic).

  • Line 229–230: “We then confirmed the covariates... as selected covariates...”
      • Clarify how covariates were selected (e.g., p-value cutoff, prior knowledge)

Logistic Regression & Effect Modification by UPF Intake

  • Line 238–239: The unadjusted OR for high UPF intake is <1 (e.g., 0.370), indicating lower odds of diabetes.
      • Recommendation: Include a brief explanation. Readers may question this result if not addressed here.

  • Lines 252–255: In the interaction model, the association between depression and diabetes becomes non-significant (UPF1) or attenuated (UPF2). Readers may find this confusing.
      • Clarify that this is expected due to the inclusion of interaction terms.

Age-Stratified Analysis

  • Lines 276–280: Regression findings show significance only in non-elderly participants with high UPF intake.
      • This is the central finding of the study. Consider moving this paragraph earlier.
      • Also consider revising the language to “adults” and “older adults” instead of elderly.
  • Line 282: “higher adjusted R-squared” is mentioned but values are not reported.
    • Recommendation: Provide R² values.

  1. Discussion Section Review

Strengths:

  • The authors effectively revisit the central hypothesis: UPF modifies the relationship between depression and diabetes.
  • The discussion references multiple biological and behavioral mechanisms, including emotional eating, antidepressant use, and inflammation.
  • Age-stratified effects are well-noted, and potential public health implications are introduced.

Comments & Recommendations

Summary of Findings

  • Lines 292–296: Sentence structure is overly complex.
    • Recommendation: Break into two sentences and clarify the comparison group.
    • “Using a broader UPF definition (UPF1), the association between depression and diabetes was observed only among individuals with high UPF intake. Under the UPF2 definition, high UPF consumers had a significantly stronger depression–diabetes association than low UPF consumers.”

Pathways from Depression to Diabetes

  • Lines 305–307: “trichloroacetic acid” is a chemical not used in antidepressants. This should be tricyclic antidepressants.
    • Correction: Replace with “tricyclic antidepressants” or just list the classes: SSRIs, SNRIs, tricyclics.

  • Line 308–309: The study on fluoxetine and adolescent diabetes risk is helpful but needs better context.
    • Recommendation: Add a clause noting limitations, e.g., “...though this association may depend on duration and dosage, and is not universally observed.”

Pathways from Diabetes to Depression

  • Line 311: Revise for clarity: “Ultra-processed foods have become increasingly common in the diets of young people due to industrialized food systems and lifestyle shifts.”

  • Lines 321–326: Mentions HPA axis, SNS, and inflammation — these mechanisms are valid but needs further development.
    • Recommendation: Add a citation for the inflammatory pathway and briefly describe how UPFs may contribute (e.g., increased C-reactive protein levels, gut dysbiosis).

Broader Implications for UPF Reduction

  • Line 343: “plastic ingredients from excessive packaging” is not clear.
    • Recommendation: Use more accurate language: “...potential exposure to endocrine-disrupting chemicals, such as phthalates or bisphenol A, from UPF packaging.”

Clinical and Intervention Implications

  • Line 350: The phrase “healthy agents” is unclear.
    • Correction: Replace with “health professionals” or “public health practitioners.”

Limitations

  • Line 373–374: “This finding is inconsistent with other studies...” : Expand on this.
    • Clarify: “This may reflect reverse causality, where individuals diagnosed with diabetes adopt healthier diets, leading to lower reported UPF consumption.”

  • Lines 378–380: Mention of recall bias is appropriate; consider also adding limitations related to lack of portion size, nutrient density, and calorie information.

  1. Conclusion Section Review

Strengths:

  • Appropriately restates the key finding that UPF acts as an effect modifier in the depression–diabetes relationship.
  • Highlights that the effect is most pronounced in younger adults, a crucial and policy-relevant insight.
  • Calls for longitudinal studies, which is the correct next step given the cross-sectional nature of the current data.

Comments & Recommendations

Line 387–388: Suggested Revision to make conclusion clearer: “This study demonstrates that ultra-processed food (UPF) intake significantly modifies the relationship between depression and diabetes in a large Brazilian adult population.”

Line 389–390: Consider using “younger adults” instead of “non-elderly,” which can be awkward and less intuitive.

Line 391–392: Suggested Revision: “These findings underscore the importance of integrating dietary screening and mental health evaluation in diabetes prevention strategies, particularly for younger populations.”

Lines 392–394: Suggested Revision: “Future longitudinal studies are needed to assess causal direction and to further explore whether UPF intake serves as a moderator or potentially a mediator in the depression–diabetes relationship.”

Author Response

Comment 1: Lines 17-18: The background line does support the need for the study if this association has been previously investigated.

Response 1: We revised the background description to emphasize the research gap in the bidirectional relationship between depression and diabetes and its relationship with UPF (line 21-23).

Comment 2: Recommendation: Highlight the gap in the literature and why study disparities in age.

Response 2: We added the age relevant highlighting content in line 86.

Comment 3: Line 22: Given age is an important variable in this study, I suggest authors to include the age range of the adult population in the abstract.

Response 3: We added the age range (18-92 years) in line 27.

Comment 4: Methods: Suggest including how diet was assessed (qualitative FFQ).

Response 4: We added “assessed by qualitative FFQ” in line 28.

Comment 5: Lines 25–26: “employing two classification methods for UPF intake (UPF1 and UPF2)” — needs a little explanation. Recommendation: Add a clause to clarify, e.g., “based on different thresholds of weekly consumption.”

Response 5: We edited it to “employing two classification methods - UPF1 and UPF2 - based on different thresholds of weekly consumption, for high/low UPF intake” in line 31-32.

Comment 6: Unclear what 1 and 2 stands for, as in Line 27 authors use the term “high UPF”. Is it high in both UPF 1 and 2 or does this refer to something different? Recommend consistency among the terms and a brief explanation.’

Response 6: For clarification, we have added the statement “based on different thresholds of weekly consumption, for high/low UPF intake” in line 31-32.

Comment 7: Lines 28–29: “Adjusted models demonstrated elevated odds ratios (OR)...” —  It was a little unclear which reference group was being used and which variables were adjusted for. Recommendation: Specify the comparison (e.g., “compared to individuals with low UPF intake”) and clarify the model (e.g., adjusted for age, sex, etc.).

Response 7: we have added the description “Models adjusted by demographic characteristics, as well as meat and vegetable consumptions” in line 34-35, and “compared to those with a low UPF intake” in 37.

Comment 8: Line 34–36: “These results emphasize the critical role dietary patterns may play in mental and metabolic health…” Recommendation: Strengthen the takeaway: “These findings suggest that high UPF intake may influence the relationship between depression and diabetes, especially in younger adults.”

Response 8: We have revised and added lines 41-42 based on your comment. We agree that a more straightforward conclusion is more appropriate for the abstract.

Comment 9: The introduction is lacking literature on age-related disparities in terms of diet, and why it could be a variable to examine in the association between depression and diabetes modified by UPF.

Response 9: We have clarified the relationship between age and UPF in a separate paragraph in lines 86-98 and added specific references. As for the relationship between UPF and comorbidity, we first presented evidence describing the relationship between UPF and depression and diabetes, respectively, as well as age differences, and then explained the research gaps in relation to comorbidity to enhance our rational and the necessity of a new study.

Comment 10: Line 43: “Depression epidemic deserves more attention in the post-pandemic era...” Recommendation: Rephrase: “The global burden of depression has grown significantly in the post-pandemic era...”

Response 10: We appreciate the reviewer's suggestions for improving our details and follow the recommendation.

Comment 11: Line 66: “UPF originally refer to foods containing a variety of unnatural additives.” Comment: I suggest using the most recent definition here. Include a citation for the reference. For example: “Ultra-processed foods (UPFs), as defined by the NOVA classification, are industrial formulations typically containing additives, colorants, and emulsifiers not commonly found in home cooking.” Please include citations in all sentences from line 66-72.

Response 11: From lines 73 to 75 in the new version, we have explained the NOVA classification and added a new reference (reference 13) to to clarify the definition of UPF.

Comment 12: Line 73-74: Reconsider the use of “While” in the beginning of the sentence, it does not read well. Perhaps shorten and combine the sentence with the previous one.

Response 12: We adjusted the two sentences beginning with ‘While’ in lines 64 and 78 of the new version.

Comment 13: Lines 84–85: “Since the 1990s, the intake of UPFs in Brazil has steadily increased.” Recommendation: Add a supporting statistic or trend line if possible (e.g., percentage increase in caloric contribution).

Response 13: We added ‘contributing from 9% to more than 20% of total energy intake currently’ in lines 101.

Comment 14: Lines 86–89: “Using Brazilian nationally representative data, we investigated the association between depression and diabetes in Brazil and how ultra-processed food diet affect this relationship.” Recommendation: Revise: “This study uses nationally representative data to investigate whether ultra-processed food intake modifies the relationship between depression and diabetes in Brazilian adults.”

Response 14: We appreciate the reviewer's recommendations for grammar details and follow the recommendation.

Comment 15:

  • Line 98–99: The phrase “participants’ responses, about their anthropometrics, noncommunicable diseases, and lifestyle habits...”. Recommendation: Rephrase to: “This survey collected data on anthropometric measures, noncommunicable diseases, and lifestyle behaviors.”
  • The authors mention here and in the abstract that participants with complete anthropometric data were retained. However, it is unclear why anthropometric was used as the eligibility criteria. Furthermore, this information is not included in Figure 1. Please include the rationale for selecting participants with complete anthropometrics given this is a covariate.
  • Figure 1: suggest saying “missing” rather than “lacking”. Why use the UPF variable in one box and food intake in the second excluded box?

Response 15:

  • We appreciate the author's revisions to the details of the text.
  • We clarified the basic criteria for unadjusted models and the additional requirements for inclusion in adjusted models in line 144-150.
  • We deleted the “with complete anthropometric information” in line 27 since they are only applied to adjusted model.
  • We added ‘to ensure BMI could be calculated’ in line 148 to justify anthropometry as an inclusion criterion.
  • We have changed ‘lacking’ to ‘missing’ for Figure 1.
  • Food intake includes ‘animal-based foods,’ ‘plant-based foods,’ and ‘ultra-processed foods,’ so we summarized it as food intake in the adjusted model, while the unadjusted model only included UPF.

Comment 16: Line 114: Typo in “We taken weight into account…” Recommendation: “We took weights into account…”

Response 16: Typo corrected.

Comment 17: Comment: Consider explaining how the weights were incorporated (e.g., “weights were applied using Stata’s svyset command”).

Response 17: We added “by using Stata’s svyset command” to line 142.

Comment 18: Line 132: I believe there is a copy/paste error – do authors mean ‘Effect modifier: Frequency of UPF intake’?

Response 18: Thank you very much—this was indeed a copy/paste error, and the corrected subtitle has been applied in line 163.

Comment 19: Line 135: please include number of food items, whether this was a qualitative or semi-quantitative FFQ (it seems like it is a qualitative), and the timeframe for responses (past week?).

Response 19: We included the number of food items in line 174-176.We added the “qualitative” in line 166, the timeframe is actually “usually” as stated in line 168.

Comment 20: Lines 142–145: UPF classification includes soda and artificial juice, which is common in NOVA. Consider explicitly citing NOVA here for transparency.

Response 20: We cited reference 13 in line 177 to refer to the NOVA definition of UPF.

Comment 21: Include the definitions of UPF1 and UPF2.

Response 21: We added “Specifically, UPF1 defines high consumption as any UPF having a weekly consumption frequency higher than the median value for the population, while UPF2 defines high consumption as all UPFs having weekly consumption frequencies higher than the median value for the population” in line 183-186.

Comment 22: What is the oldest age of the participants in this study? From Table 1 it seems like about 75 y/o. I ask here because I wonder if authors considered a different obesity cut-off for older adults (> 65 y/o)?

Response 22: We did not apply an upper age limit in our analysis, and participants included in the study ranged up to 92 years old. We acknowledge that age-specific criteria for defining obesity could be considered, while we opted not to use a different threshold at 65 years of age. In Brazil, the legally recognized age for defining older adults is 60 years, and applying a different cutoff could introduce inconsistency and confusion in the interpretation of age-related variables. Moreover, the reference cited by the reviewer is based on data from the Mexican population, which may not be directly applicable to the Brazilian context due to important differences in racial and ethnic admixture composition, lifestyle factors, and geographic locations, and major population in Brazil are not Hispanic. For these reasons, we maintained a consistent age threshold aligned with Brazilian legal and demographic standards.

Comment 23: Please include the rationale for using logistic regression models that estimate odds ratio rather than a model to estimate prevalence ratio (e.g. log-binomial models).

Response 23: We used multivariate logistic regression models to estimate odds ratios (ORs) and 95% confidence intervals for the association between depression, UPF intake, and diabetes. Although alternative approaches such as log-binomial models can directly estimate prevalence ratios (PRs), we selected logistic regression due to its greater model stability and convergence properties, particularly in large, complex survey data with multiple interaction terms. Additionally, since the outcome (diabetes) had a prevalence below 10% in our analytic sample, the odds ratio closely approximates the prevalence ratio in this context.

Comment 24: Did authors explore a three-way interaction term between depression*UPF*age before stratifying by age? Please describe and include the p for this 3-way interaction to help support the stratified analysis.

Response 24: We added the result at line 245-249: “We tested a three-way interaction between depression, UPF intake, and age in the fully adjusted model. The interaction term was statistically significant (p = 0.037), indicating that the modifying effect of UPF intake on the depression-diabetes association differs across age groups.”

Comment 25: Please include range of age of participants

Response 25: We added “ranging from 18 to 92 years” in line 273-274.

Comment 26: Table 1 is not stand-alone as is. Please include n and % of diabetes/non-diabetes, definition of UPF 1 and 2, describe abbreviations as footnote.

Response 26: We edited the sub-heading and footnotes based on your kindly suggestions.

Comment 27: Line 218: “Physically active participants comprised 33.42%...” Add a clarifying statement, e.g., “...based on self-reported activity ≥150 minutes per week.”

Response 27: We added “based on self-reported activity ≥150 minutes per week” in line 270.

Comment 28: Lines 224–227: The association findings (e.g., “higher prevalence of diabetes among those physically inactive, obese, or with depression...”) are meaningful but buried in a long sentence. Recommendation: Consider separating these associations by variable group (e.g., behavioral vs. sociodemographic).

Response 28: We split the sentence into 2 sentences. Now they are “For health-related status and behaviors, individuals with a higher prevalence of diabetes in our sample were more likely to report depression, obesity, and low UPF consumption, as well as higher intake of plant-based whole foods.” for the key factors in our manuscript, the next sentence is related with other behaviors:  “They were also more likely to have ever smoked, be physically inactive, and not have consumed alcohol in the past month. The average age of people with diabetes was higher than that of people without diabetes.” in line 276-281.

Comment 29: Line 229–230: “We then confirmed the covariates... as selected covariates...” Clarify how covariates were selected (e.g., p-value cutoff, prior knowledge)

Response 29: We added “based on the p-value cutoff point at 0.05” in line 283.

Comment 30: Line 238–239: The unadjusted OR for high UPF intake is <1 (e.g., 0.370), indicating lower odds of diabetes. Recommendation: Include a brief explanation. Readers may question this result if not addressed here.

Response 30: We added “This inverse association in the unadjusted model likely reflects reverse causality, whereby individuals previously diagnosed with diabetes may have reduced their UPF intake as part of dietary management recommendations.” in line 301-304.

Comment 31: Lines 252–255: In the interaction model, the association between depression and diabetes becomes non-significant (UPF1) or attenuated (UPF2). Readers may find this confusing. Clarify that this is expected due to the inclusion of interaction terms.

Response 31: We added “which is expected due to the inclusion of interaction terms” in line 316.

Comment 32: Lines 276–280: Regression findings show significance only in non-elderly participants with high UPF intake. This is the central finding of the study. Consider moving this paragraph earlier. Also consider revising the language to “adults” and “older adults” instead of elderly.

Response 32: Since our overall writing structure is descriptive analysis: regression analysis-stratification, we believe that a step-by-step presentation of results is necessary. We have splitted the adult population in: ‘adults’ and “older adults” throughout the manuscript.

Comment 33: Line 282: “higher adjusted R-squared” is mentioned but values are not reported. Recommendation: Provide R² values.

Response 33: We added R2 details in table 2.

Comment 34: Lines 292–296: Sentence structure is overly complex. Recommendation: Break into two sentences and clarify the comparison group. “Using a broader UPF definition (UPF1), the association between depression and diabetes was observed only among individuals with high UPF intake. Under the UPF2 definition, high UPF consumers had a significantly stronger depression–diabetes association than low UPF consumers.”

Response 34: We edited and applied your kindly suggestion to line 355-358.

Comment 35: Lines 305–307: “trichloroacetic acid” is a chemical not used in antidepressants. This should be tricyclic antidepressants. Correction: Replace with “tricyclic antidepressants” or just list the classes: SSRIs, SNRIs, tricyclics.

Response 35: We made the correction in line 371.

Comment 36: Line 308–309: The study on fluoxetine and adolescent diabetes risk is helpful but needs better context. Recommendation: Add a clause noting limitations, e.g., “...though this association may depend on duration and dosage, and is not universally observed.”

Response 36: We added in line 376.

Comment 37: Line 311: Revise for clarity: “Ultra-processed foods have become increasingly common in the diets of young people due to industrialized food systems and lifestyle shifts.”

Response 37: We revised it in lines 378-379.

Comment 38: Lines 321–326: Mentions HPA axis, SNS, and inflammation — these mechanisms are valid but needs further development. Recommendation: Add a citation for the inflammatory pathway and briefly describe how UPFs may contribute (e.g., increased C-reactive protein levels, gut dysbiosis).

Response 38: We expanded our statement to include information related to C-reactive protein levels and UPF causing inflammation and increasing intestinal permeability, in line 391-395.

Comment 39: Line 343: “plastic ingredients from excessive packaging” is not clear. Recommendation: Use more accurate language: “...potential exposure to endocrine-disrupting chemicals, such as phthalates or bisphenol A, from UPF packaging.”

Response 39: We revised it in line 418-419.

Comment 40: Line 350: The phrase “healthy agents” is unclear. Correction: Replace with “health professionals” or “public health practitioners.”

Response 40: We changed them into “health professionals” in line 430 and 472.

Comment 41: Line 373–374: “This finding is inconsistent with other studies...” : Expand on this. Clarify: “This may reflect reverse causality, where individuals diagnosed with diabetes adopt healthier diets, leading to lower reported UPF consumption.”

Response 41: We revised it in line 452-453.

Comment 42: Lines 378–380: Mention of recall bias is appropriate; consider also adding limitations related to lack of portion size, nutrient density, and calorie information.

Response 42: We mentioned it now in line 458.

Comment 43: Line 387–388: Suggested Revision to make conclusion clearer: “This study demonstrates that ultra-processed food (UPF) intake significantly modifies the relationship between depression and diabetes in a large Brazilian adult population.”

Response 43: We changed the staring of the conclusion in line 466 - 468.

Comment 44: Line 389–390: Consider using “younger adults” instead of “non-elderly,” which can be awkward and less intuitive.

Response 44: As we mentioned in Response 33, we have changed the definitions of adults as ‘adults’ and “older adults” throughout the manuscript.

Comment 45: Line 391–392: Suggested Revision: “These findings underscore the importance of integrating dietary screening and mental health evaluation in diabetes prevention strategies, particularly for younger populations.”

Response 45: We have made changes based on your suggestions and those of other reviewers in line 469-471.

Comment 46: Lines 392–394: Suggested Revision: “Future longitudinal studies are needed to assess causal direction and to further explore whether UPF intake serves as a moderator or potentially a mediator in the depression–diabetes relationship.”

Response 46: We accepted your kindly suggestion in 483-485.

Round 2

Reviewer 3 Report

Comments and Suggestions for Authors

Thanks for addressing all my comments - the paper has certainly improved.